# Investigation of Leukocyte Viability and Damage in Spiral Microchannel and Contraction-Expansion Array

**DOI:** 10.3390/mi10110772

**Published:** 2019-11-12

**Authors:** Thammawit Suwannaphan, Werayut Srituravanich, Achariya Sailasuta, Prapruddee Piyaviriyakul, Suchaya Bhanpattanakul, Wutthinan Jeamsaksiri, Witsaroot Sripumkhai, Alongkorn Pimpin

**Affiliations:** 1Department of Mechanical Engineering, Faculty of Engineering, Chulalongkorn University, Bangkok 10330, Thailand; thammawit.su@student.chula.ac.th (T.S.); werayut.s@chula.ac.th (W.S.); 2Department of Pathology, Faculty of Veterinary Science, Chulalongkorn University, Bangkok 10330, Thailand; achariya.sa@chula.ac.th; 3Department of Physiology, Faculty of Veterinary Science, Chulalongkorn University, Bangkok 10330, Thailand; prapruddee.p@chula.ac.th (P.P.); sudchaya.bha@student.chula.ac.th (S.B.); 4Thailand Microelectronic Centre, Ministry of Science and Technology, Chachoengsao 24000, Thailand; wutthinan.jeamsaksiri@nectec.or.th (W.J.); witsaroot.sripumkhai@nectec.or.th (W.S.)

**Keywords:** microfluidics, cell viability, cell morphology, intracellular structures, Leukocytes, spiral microchannel, contraction-expansion array

## Abstract

Inertial separation techniques in a microfluidic system have been widely employed in the field of medical diagnosis for a long time. Despite no requirement of external forces, it requires strong hydrodynamic forces that could potentially cause cell damage or loss during the separation process. This might lead to the wrong interpretation of laboratory results since the change of structures and functional characteristics of cells due to the hydrodynamic forces that occur are not taken into account. Therefore, it is important to investigate the cell viability and damage along with the separation efficacy of the device in the design process. In this study, two inertial separation techniques—spiral microchannel and contraction-expansion array (CEA)—were examined to evaluate cell viability, morphology and intracellular structures using a trypan blue assay (TB), Scanning Electron Microscopy (SEM) and Wright-Giemsa stain. We discovered that cell loss was not significantly found in a feeding system, i.e., syringe, needle and tube, but mostly occurred in the inertial separation devices while the change of cell morphology and intracellular structures were found in the feeding system and inertial separation devices. Furthermore, percentage of cell loss was not significant in both devices (7–10%). However, the change of cell morphology was considerably increased (30%) in spiral microchannel (shear stress dominated) rather than in CEA (12%). In contrast, the disruption of intracellular structures was increased (14%) in CEA (extensional and shear stress dominated equally) rather than spiral microchannel (2%). In these experiments, leukocytes of canine were used as samples because their sizes are varied in a range between 7–12 µm, and they are commonly used as a biomarker in many clinical and medical applications.

## 1. Introduction

Over the past decades, microfluidic devices have been developed for several purposes and specific applications, such as cell’s biological study [1,2], drug delivery [3], disease diagnosis [4] and cell manipulation [5]. Among cell manipulation techniques, inertial microfluidics is a passive technique that functions without applying external forces and requires only the movement of fluid. This makes the system simple and suitable for high throughput analysis [5,6]; these advantages have drawn great attention in both research and clinical applications, as they hold a promise for cell sorting.

Generally, there are four basic types of microchannel structures used in inertial principle. In each type, it might be suitable for different purposes of specific applications. They are: (1) straight channel with circular [7], square [8] and rectangular cross-section [9]; (2) serpentine channel with asymmetric [10] and symmetric shape [11]; (3) spiral microchannel with various shapes of cross-sections [12], double spiral [13] and a multiplexed spiral [14]; and (4) contraction-expansion array (CEA) with a diamond-shaped chamber [15] and a multi-orifice microchannel [16]. Among these techniques, we focused on spiral microchannel and contraction-expansion array (CEA), because both types are able to separate cells/micro particles more than two different sizes and provide a high efficacy of cell separation.

Spiral microchannel is a size-based passive sorting mainly depending on magnitude of hydrodynamic force, shape of cross-section and size of cells/micro particles [17,18]. The working principle of this technique is that cells/micro particles are experienced by two forces in opposite directions such as lift force and Dean drag force (see Figure 1a). As for lift force, it is the summation of two forces such as shear-induced lift force and wall-induced lift force. The Dean drag force is generated by the secondary flow induced on the curvature of a spiral microchannel. These forces help focusing the path-line of particles apart from the ones with different sizes. In general, the bigger cells/micro particles move towards the inner wall and the smaller cells/micro particles move towards the outer wall. However, the equilibrium position in the lateral direction on the channel cross-section does not always happen in all cases, depending on many factors such as the curvature ratio, flow rates and size of cell/particles.

Contraction-expansion array is another inertial approach that relies on size-based differential equilibrium position of the cells/micro particles [19,20]. There are two forces, namely inertial lift force and Dean drag force, that are generated by induced vortices in the transition region between the expansion and contraction of the microchannel. Briefly, once cells/micro particles move downstream of the straight microchannel, they migrate across streamlines and equilibrate at the equilibrium position in the lateral direction (near the side wall). When they enter the expansion region, the flow streamline bends due to decelerated flow according to the increment of cross-section. After that, the flow is accelerated, creating secondary flow (Dean flow) in the form of two counter rotating vortices close to the entrance of the contraction region. Due to a smaller space of the contraction, this region yields a higher shear rate, leading to large lift forces that allows particles to laterally move across the streamlines. Large particles migrate across the streamline faster than smaller ones, so that there is a higher possibility to cross the boundary streamline between the main and sheath flows. If the particles move laterally long enough, they are entrained into the sheath flow and flow towards the side outlet in the expansion. Simultaneously, small particles exit through the main outlet (see Figure 1b).

According to the studies of flow in pipe in fluid mechanics, the highest shear stress or wall-shear stress is the shear stress in the layer of fluid next to the wall of a pipe. Likewise, highest shear stress in spiral microchannel is exerted close to the wall of microchannel. In contrast, extensional stress is the normal stress exerted by the abrupt change in cross-sectional area of contraction-expansion array. These two stresses are the hydrodynamic forces per unit area that are developed by injecting fluid flow in the microfluidic device.

Despite that, cells inevitable encounter stresses caused by hydrodynamic forces resulting in either cell damage or loss since it requires high flow rate to create adequate magnitude of hydrodynamic forces. The study of cell loss, i.e., damage or death and hydrodynamic force is rare, and few experiments have been conducted in terms of magnitude of hydrodynamic force and percentage of cell loss. One possible explanation for cell loss is due to the high stresses in fluid flow—shear and extensional stresses—which should also be considered in the study of cell sorting.

### 1.1. The Studies of Stresses

Various types of cell have been studied with different techniques to investigate the effects of shear stresses thus far. Nevaril et al. and Leverett et al. investigated the effects of hemolysis of red blood cells (RBC). They discovered that magnitude of stresses caused cells to be ruptured severely around 150–300 Pa [21,22]. On the other hand, Sutera and Mehrjardi studied turbulent Couette flow and reported that increasing shear stress and exposure time resulting in the deformation of RBCs and also become fragmented [23]. Hybridoma cell is another sample that has been tested for many researches. Smith et al. demonstrated the investigation of shear sensitivity including cell viability and morphology on mouse hybridoma cells using a Couette viscometer for 15 h. The results indicated that cell damage was observed, while cell viability was reduced from 85% to 70% with a shear stress of 0.67 Pa; however, the change of cell morphology was not found at this order of force magnitude [24]. The study of Petersen et al. reported the order of stress affecting viability of hybridoma cells from 0 to 5 Pa with exposure time from 0 to 10 min [25]. Similarly, Schürch et al. investigated shear stress on hybridoma cells using a rotating viscometer. The results indicated that the loss of cell viability started to appear at the shear stress of less than 6 Pa [26]. Abu-Reesh and Kargi experimented with both laminar and turbulent shear stress affecting cell viability and at different levels and exposure time. Above the threshold of shear stress at 5 Pa in turbulent flow resulted in 50% loss of cell viability [27].

As mention above, although some studies of shear stress used the same type of cells in their research, the examined cells in different approaches would withstand shear stress in different order and respond to shear stress in different way. Therefore, the conclusion may not be made because the data of shear stress affecting on cells still have been limited and relatively had small numbers of applications to be proved. In addition, in the past, the studies of shear stress were intensively focused. However, nowadays, the studies of extensional stress are attracted attentions to the fact that extensional stress could potentially be the main cause of cell loss instead of shear stress.

For the extensional stresses, the situation is similar to that in the study of shear stress. The study of Tanzeglock et al. (2009) was one of the most first reviews of shear and extensional stress affecting on cells using rheometer and syringe. They discovered that extensional stress about 1.09 Pa caused cells to become apoptosis death for Human embryonic kidney (HEK) cells and 3.05 Pa for Chinese hamster ovary (CHO) cells while necrosis prevailed around 1 Pa in shear flow and 500 Pa in extensional flow [28]. Down et al. confirmed that extensional stress was developed at the capillary entrance. The results indicated that extensional stress may have a significant factor in contributing to failure of the RBC membrane and the damage threshold for hemolysis was about 3000 Pa, which can cause RBC trauma [29]. The study of Aguado et al. suggested that two factors that had the effect on cell loss were shear and extensional stress, especially extensional flow caused mechanical disruption in a syringe needle [30]. Yen et al. (2015) studied the effects of shear and extensional stress acting as the force behind hemolysis. The study suggested that extensional stress played a significant role in cell damage rather than shear stress and the threshold value of extensional stress was about 1000 Pa with the exposure time about 0.06 ms [31]. Bae et al. (2016) demonstrated the effects of extensional stress with Chinese hamster ovary (CHO) cells. CHOs were introduced and experienced extensional stress at the middle of microfluidic cross-slot geometry channel. They discovered that the critical extensional stress that could damage cell mechanically was about 250 Pa [32].

As can be seen from the above-mentioned studies, cells have several ways to respond to stress levels–physical and biological changes. The effects of cellular responses could be various, such as cell deformation, injury, lysis and loss depending on cell types, types of stresses, approaches, magnitude of stress and exposure time. However, cell types are limited, and the results with different measurement techniques lead to different experimental results. Furthermore, cellular heterogeneity also plays a significant role in different experimental results. Consequently, the above data cannot be clearly concluded due to insufficient data of stresses affecting cells. To have a better understanding of stress on cells, we summarized the data of stress on cells including methods, types of cells, types of stress, magnitude of stress and exposure time (see in Table 1).

### 1.2. Cellular Stress Responses

Loss of viability or cell loss is related to cell lysis because it is the most extreme response of cells to an imposed hydrodynamic stress [33]. Cell loss takes many forms, including apoptosis and necrosis. Cell loss could be observed as shrinkage, blebbing of cells, uptake of water, swelling and cellular lysis [34]. The experiment by Tanzeglock et al. also showed that the order of 0.59 Pa did not cause HEK cells to die. However, necrosis and permanent deformation of HEK cells occurred to cells when hydrodynamic stress was 2.09 Pa [28].

Membrane integrity, it is an indicator that can evaluate the loss of viability as well. One of the most commonly approaches of measuring membrane integrity is a trypan blue dye exclusion test. The numbers of viable and dead cells are measured microscopically using a hemocytometer because it is easy to use and able to measure dead cells immediately. Therefore, trypan blue dye exclusion test becomes our primary method to examine the loss of viability in this work.

Morphological variation is one of the indicators in terms of the change in cell size and deformation that are associated with the hydrodynamic environment. Tanzeglock et al. examined the shear and extensional stress on cells measuring morphological changes of cells—decrease in cell size and increase in cell granularity [28]. Similarly, an experiment by Stathopoulos and Hellums observed and measured the change in cell morphology of Human embryonic kidney cells [35].

As can be seen from the above studies, the diversity of factors including cell types, experimental approaches, magnitude of stress, types of hydrodynamic stress, exposure time/area, cellular heterogeneity and cellular stress responses lead to different experimental results.

In our previous study, both spiral microchannel and CEA were examined with Mast cell tumor (MCT) cells, Leukocytes and Madin-Darby Canine Kidney (MDCK) cells. The main reason that we used both devices was because they are inertial microfluidic techniques, only use a hydrodynamic approach to sort particles/cells, do not require external forces that may affect cell biological property and provide a high efficiency of cell separation. However, the results showed that some cells were loss during the separation process. This unexpected result encouraged us to investigate cell loss and damage in our sorting devices.

From the engineering point of view, it is significantly important to examine our devices and investigate shear and extensional stress that could affect cell viability, cell morphology and intracellular structures in the setup of spiral microchannel and CEA (see Figure 1). The experiment includes a feeding system, i.e., syringe, needle, a silicone tube and a cell sorter, i.e., spiral and CEA part. In this work, three methods including a trypan blue assay, Scanning Electron Microscopy (SEM) and Wright-Giemsa staining under various conditions were employed. To correlate the cell viability and cell damage with the hydrodynamic stresses, the experiments were conducted together with computational simulation to explain some flow phenomena that happened in each component in spiral microchannel and CEA device.

## 2. Experiments

### 2.1. Test Methods

Whole blood (Canine) was acquired within one day of collection at Chulalongkorn University, Small-Animal hospital (Bangkok, Thailand). The cell preparation protocols were started with collecting canine blood using a pipette and mixed 3 mL of whole blood with 45 mL of RBC lysing buffer into a 50 mL plastic centrifuge tube. The whole blood and lysis buffer were mixed well together using a biomixer for 5 min and then spun down at 3000 rpm and 4 °C for 20 min using centrifugation. We adapted the new protocols that could reduce the steps of washing that could reduce a risk of cell damage from centrifugation process using a pipette washing leukocytes with Phosphate-buffered saline (PBS) manually inside the plastic centrifuge tube to remove lysis buffer. They were then washed again by filling to a 50 mL plastic centrifuge tube with PBS and then centrifuge. However, this technique might loss some leukocytes by washing manually, but it is significantly useful to prevent most leukocytes from the damage of centrifugation. This protocol took about one hour with >95% cell viability and ready to be used for testing. In this experiment, the initiated concentration was diluted down to 1 × 10^7^ cells/mL for all experiments. It should be noted that our sample is the mix of donated blood from healthy and unhealthy dogs. To control the quality of sample, it will not be used in our experiment if cell viability of leukocyte is lower than 90%.

For the cell viability measurement, Trypan Blue assay (TB) (Hyclone^TM^, GE Healthcare Life Sciences, Chicago, IL, USA) was used, since it is the gold standard for identification of the dead/alive cells. To verify cell death, Trypan Blue assay was performed under the standard protocols to test cell viability [36]. The numbers of dead cells are visually inspected and enumerated using a hemocytometer (Blood Counting Chamber Bright Line and Cover glass) (Boeco, Hamburg, Germany). A dead cell appears as a dark feature with a circle, whereas alive cell appears as a bright feature with a circle. Cell suspension after the preparation is set aside as control condition. It showed that the percentage of leukocyte viability of the controls were about 96%. In this work, the samples were counted in three sets and examined three times for each condition. The percentage of cell viability was then calculated. The total number of cells counted for the trypan blue test was about 35,000 cells.

For the cell deformation measurement, Scanning Electron Microscopy (SEM) (JSM-IT-500HR, JEOL Ltd., Tokyo, Japan) is used for investigating of cell morphology. The SEM protocol was started with preparing samples for Scanning Electron Microscopy using leukocyte suspension obtained from whole blood. One drop (50 μL) of leukocyte suspension was gently placed with a pipette on a cover slip and left for 2 min at 25 °C. It is noted that the suspension should be all over the plate. Specimens were fixed with 2.5% glutaraldehyde in 0.1 M phosphate buffer pH 7.2 for 1 h. Specimens were washed twice with phosphate buffer to remove glutaraldehyde, and then once with distilled water for 5 min/each. Dehydrating specimen with a graded series of ethanol (30%, 50%, 70%, 95% 5 min/each and 100% three times, 5 min/time, respectively). Fluid was removed from the specimens by evaporating with high-pressure heating to the critical point dryer (Leica model EM CPD300, Austria). The specimen was mounted onto stubs with conductive tape and coat with gold (sputter coater, Balzers model SCD 040, Germany). Finally, they were observed by means of SEM [37]. In this study, we define “normal cell” to refer to cells in a circular shape. For the term “degenerated cells” refers to cell deformation, membrane damage and nucleus stretching (see Figure 2a). Six images were taken from four sets in each condition, and the numbers of normal cells and degenerated cells were counted. In this work, the samples were counted in three sets and examined three times. The total numbers of cells for SEM investigation were about 33,000 cells.

For the intracellular structure measurement, Wright-Giemsa stain is a tool that allowed us to observe intracellular structures including cell membrane, cytoplasm and nucleus. The Wright-Giemsa stain protocol was started with smearing thoroughly leukocytes on glass slide and completely dried with dry air without heat. Placing Wright-Giemsa stain in a Coplin jar and filled another Coplin jar with phosphate buffer. Fixing smears in absolute methanol for 10–15 s and completely dried with dry air again. Staining smears in Wright-Giemsa stain solution for 10–15 s and rinsed stained smears with Phosphate Buffer for 10–15 s and dried the smears with dry air. Permanent mounts can be made. Finally, examining stained smears under a microscope. In order to avoid any confusion of interpretation, here, “normal cells” is used to refer to cells that are in a circular shape, and consist of three main structures—cell membrane, cytoplasm and nucleus. The term “degenerated cells” refers to cells that miss one of the three main structures (see Figure 2b). In the tests, leukocytes were spread thinly onto a glass slide, and then, 100 cells were randomly chosen and classified as normal or degenerated cells under a light microscope. The samples were counted in three sets and examined three times. The total number of leukocytes for Wright-Giemsa Stain investigation was about 17,000 cells.

### 2.2. System Fabrication

For silicon mold fabrication, the process began with a silicon mold. A 6-inch silicon wafer was cleaned by dipping it in a piranha solution to remove organic contaminations. Then, the silicon mold was spin-coated with the Hexamethyldisilazane (HMDS) and baked at 90 °C for 90 s to improve the photoresist, PR (Soho Sumitomo PFI27C9), adhesion to the oxidized silicon wafer surface. The photoresist was spin-coated at 1000 rpm for 20 s to obtain 2 µm thickness. The pattern of spiral microchannel/CEA was constructed by photolithography method using a mask aligner (EVG 620, EV Group, Oberosterreich, Austria). The spin-coated wafer was exposed with UV-light through a lithograph mask for 5 s. After that, the silicon wafer was baked at 110 °C for 100 s and then developed with SD-W for 75 s, and hard baked at 120 °C for 80 s. Finally, the wafer was etched using Deep reactive-ion etching (DRIE) technique (SF6/C4F8 gases) to create the pattern of microchannels to the desired depth.

Polydimethylsiloxane, PDMS (Sylgard 184 Silicone Elastomer, Dow Corning, Auburn, MI, USA) and a curing agent were mixed at the ratio of 10 to 1 by weight. PDMS structures were casted by pouring on the silicon mold. After that, PDMS was cured in an oven at 75 °C for 90 min to form a spiral microchannel/CEA, and then peeled off. Finally, inlet and outlet reservoirs were punched using a pin vise to connect silicone tubes. For the device assembly, the PDMS channel and glass microscope slide were exposed to oxygen plasma under 40 sccm of O_2_ with 30 watts for 90 s and put in contact with one another to enclose the spiral microchannel/CEA chip.

### 2.3. Feeding System and Test Conditions

Generally, the experiments of all cases were set (see Figure 3a). The sample was loaded in the feeding system mounted on syringe pumps (F-100, Chemyx, Stafford, TA, USA) and injected through the setup of the devices. Furthermore, separation process was microscopically observed and recorded in a computer and then the samples were collected in microtubes to further investigation. In our investigation, the feeding system was examined without the sorting device.

One of the biggest challenges is to maintain and prolong cell viability on the site of delivery. Therefore, the possibility of cell damage or loss in feeding system including syringe, needle and silicone tube must be investigated initially. In this test, three different sizes of syringe—1, 2.5 and 5 mL—were tested with a length of 59.89 mm and an inner barrel’s diameter of 4.61, 7.29 and 10.30 mm attached to a 22 G needle with an inner diameter of 0.41 mm and a length of 1.5 cm. The tip of needle was also connected to a tube (Clear C-FLEX Tubing, Cole-Parmer Instrument Company, Vernon hills, IL, USA) with a length of 20 cm and with an outer and inner diameter of 0.72 and 0.41 mm. The feeding system was examined at various flow rates: 0, 0.3, 1, 5 and 8 mL/min (see Figure 3b).

### 2.4. Spiral/Contraction-Expansion Array (CEA) Systems and Test Conditions

Spiral microchannel was designed as an Archimedean spiral microchannel, consisting of five loops with two inlets. The width of 500 μm and the height of 130 μm were fixed with the average radius of curvature of 0.75 cm and increasing at a constant rate of angle. The outlet was divided into two parts; the straight channel with a length of 10 mm with a gradual expansion connected to the 10 outlets (see Figure 3c). Each outlet was designed with a width of 200 μm at the same height. The experiment was examined at two conditions: 2 and 10 mL/min. 2 mL/min (1 mL/min at each inlet) was the optimum condition that provided the highest efficacy of cell separation (10, 15 and 20 µm in diameter). At 10 mL/min (5 mL/min at each inlet), the spiral chip could withstand this flow rate. The samples were collected in microtubes at the 10 outlets after flowing through the complete setup.

For the CEA setup, there are two contraction-expansion chambers, namely the upstream and downstream microchamber. The CEA consisted of the square channel of 50 μm connected to the expansion chamber downstream with the width of 500 μm (see Figure 3d). On both sides of the upstream microchamber, there are two side outlets connected to the downstream microchamber to sort cells with a higher efficacy and purity. The experiment was examined at 0.3 mL/min in the main and 0.24 mL/min in the buffer inlet. These flow conditions were the optimal condition with which the system could separate the cells of 10, 15 and 20 µm with a high efficacy. Similar to the other cases, the samples were also collected in microtubes after flowing through the complete setup.

## 3. Simulation Study

A computational simulation based on finite element method was employed with COMSOL Multiphysics® version 5.3a (COMSOL, Inc., Los Altos, CA, USA) to analyze flow phenomena—the velocity field, stress tensors and path of streamlines—in each part of microfluidic devices, including the feeding system, spiral microchannel and CEA.

### 3.1. Geometry and Boundary Conditions of the Feeding System

The geometry of the syringe was simulated from the actual syringe with the volume of 1, 2.5 and 5 mL with a straight needle of 22 G. The length of syringe barrel was kept constant in all designs and the angle between the syringe’s barrel and the attached needle was also kept at 90°. To ensure that all boundary and conditions were correct and proper, this simulation was compared with the simulation of Down et al. and Yen et al. [29,31]. Two-dimensional axisymmetric simulation was used to reduce computational time. Geometry of syringe was designed as two rectangles unionized and revolved to build barrel’s geometry and needle. Corner refinement and distribution were also employed in specific areas to provide more accurate data, especially at the corner between barrel and attached needle. The inlet flow was defined as a uniform flow; normal inflow velocity was chosen at the inlet with 0.2 mm/s (for 1 mL/min) and 1.6 mm/s (for 8 mL/min). At the outlet, pressure at the cross section was uniform; suppress backflow were assumed. In the preliminary simulation, grid dependency analysis based on the maximum stress at the corner was implemented. However, the leukocyte was about 7–12 µm in diameter; thus, it was decided to minimize the size of mesh (<1 µm) at the corner where the maximum extensional and shear stress occurred (see Figure 4a). Finally, the maximum and minimum element sizes for the feeding system were 10.6 and 1 µm, respectively. The final number of meshes of the feeding system was about 20 million meshes.

### 3.2. Geometry and Boundary Conditions of Spiral Microchannel

The actual spiral microchannel was designed with the width and height of about 500 and 130 µm with the average radius of curvature of 0.75 cm increasing at a constant rate of angle with five loops consisting of a straight channel and 10 outlets. However, due to the computational limitation and it being time consuming, only one loop spiral was simulated. The flow inlet was defined as a uniform flow of which the normal inflow velocity was set to 0.5 m/s (for 2 mL/min) and 2.5 m/s (for 10 mL/min). At the outlet, pressure at the cross section was uniform and suppress backflow were selected. Grid dependency analysis based on the maximum extensional and shear stress in the middle of spiral microchannel was employed (see Figure 4b). Finally, the maximum and minimum element sizes for spiral microchannel were 13.90 and 0.91 µm. The final number of meshes of the spiral microchannel was about 6.3 million meshes.

### 3.3. Geometry and Boundary Conditions of the Straight Channel and Outlets

The outlets were modeled into two parts; the 10 mm straight channel followed by the expansion with a length of 10 mm and expanded angle of 9° before splitting into five symmetry outlet sub-channels. A uniform flow was introduced at the inlet of 2 mL/min for optimum condition and 10 m/s for maximum stress condition. Suppress backflow were selected at the outlet. Conner refinement was employed at all corners of the model. Grid dependency analysis based on the maximum extensional and shear stress was employed at the expanded area (see Figure 4c). The maximum and minimum element sizes for this model were 25 and 2.5 µm. The final number of meshes of the spiral microchannel was about 2.5 million meshes.

### 3.4. Geometry and Boundary Conditions of CEA

For CEA, only the upstream microchamber that had faster flow rate was simulated as symmetry in x-y and x-z axis with 250 and 25 µm of microchamber. There are two outlets such as main and side outlets. The main inlet was defined as a uniform flow which normal inflow velocity was chosen at 2 m/s (for 0.3 mL/min). For side outlet, we defined the main inlet at 0.3 mL/min and then further increased in the length of side outlet while calculated the appropriate ratio of flow rate in the main and the side outlet. The length of the side outlet was chosen at the flow rate of 0.02 mL/min corresponding to the optimum condition in our experiment. Similar to the feeding system simulation, grid dependency analysis based on the maximum extensional and shear stress was employed at the abrupt change in cross section between contraction-expansion in upstream microchamber (see Figure 4d). The maximum and minimum element sizes were 6.225 and 0.096 µm for all domains of CEA. The final number of meshes of CEA was about 9 million meshes.

In all cases, the medium (PBS) was assumed to be water with density of 998 kg/m^3^, viscosity of 1 mPa·s, and the flow was considered as laminar flow. The steady state for incompressible fluids was defined. In addition, there was no slip condition for the walls in all simulations.

## 4. Results

To gain insight into the effect of stresses on cells in each component, the results of each experiment will be analyzed by using a computational simulation including the evaluation of the extensional and shear components of stress tensor, exposure time and the critical areas. For these advantages of simulation, we will be able to understand the causes of cell damage and loss in each component including feeding system, spiral microchannel and CEA.

### 4.1. The Results of Feeding System

The experimental results showed that, in the feeding system, increasing the ratio of the abrupt change in cross section between barrel and nozzle by increasing the volumes of syringe of 1, 2.5 and 5 mL but the flow rate fixed at 1 mL/min, the percentage of normal cells were 95.8% (control), resulted in 94.6%, 97.9% and 94.5% of cell viability. Likewise, increasing extensional stress by increasing flow rates of 0.3, 1, 5 and 8 mL/min but the 5 mL syringe fixed, the percentage of normal cells were 96.6% (control), resulted in 96.8%, 96.7% and 96.8% cell viability. Therefore, either increasing the volumes of syringe or increasing flow rates, cell viability in these both conditions were not significantly different from the controls under a light microscopy with a trypan blue assay. The data of SEM showed that normal cells were decreased from 97.6% to 94.5%, 93.8%, 90.2% and 87.4% at 0.3, 1, 5 and 8 mL/min, respectively. This results also agreed with the study of Bea et al. [32], they found that cells get more deformed as the flow rate increases in the extensional flow. In addition to Wright-Giemsa stain test, it was found that the percentage of normal cells were decreased from 96.8% to 95.6%, 94.1%, 91.2% and 89.0% at 0.3, 1, 5 and 8 mL/min, respectively (see Figure 5).

In this study, the magnitude of stresses was represented as stress tensor. However, for ease of understanding the types of stress, we divided the stress tensor into two types of stress: total extensional components of stress tensor (the root sum squared of all extensional components of stress tensor) and total shear components of stress tensor (the root sum squared of all shear components of stress tensor).

The results showed that the magnitude of the total extensional components of the stress tensor was about 24 Pa (see Figure 6a), while total shear components of stress tensor decreased to 5 Pa at the corner (along path 1 at 1 mL/min), but exposure time was only 0.25 ms (exposure time was calculated, i.e., 2.5 µm/0.01 m/s ≈ 0.25 ms) at the corner between a barrel and needle—2.5 × 2.5 µm^2^ (for path 1) and about 0.175 ms (exposure time was calculated, i.e., 3.5 µm/0.02 m/s ≈ 0.175 s)—3.5 × 3.5 µm^2^ (for path 2). As can be seen from the stress area, highest stress area of 2.5 × 2.5 µm^2^ was smaller than the size of leukocytes. Therefore, the small area of high stresses (>24 Pa for extensional stress and >5 Pa for shear stress) may not affect cell viability after cells flowing through in this area. In addition, the needle size is around 410 µm in diameter; therefore, only small groups of cell could flow through in this high stress area.

In addition to have a better understanding of which type of stress is dominant throughout the part of device, the average total stress components of stress tensor were also calculated by averaging the root sum squared of all extensional or shear components of stress tensor along the streamline of all four paths (see Figure 6c). In this way, we will be able to understand which types of stress are dominated in each streamline. Figure 6c shows that average total extensional was higher than shear components of stress tensor for all four paths at 1 mL/min. This implied that extensional stress was dominant throughout all streamlines especially at the corner of the syringe.

In the case of increasing the flow rate to 8 mL/min, the data showed that the total extensional components of the stress tensor increased to 500 Pa at the corner, while exposure time was significantly reduced to 0.0125 ms (exposure time was calculated, i.e., 2.5 µm/0.2 m/s ≈ 0.0125 ms) (see Figure 6b) and it was higher than the total extensional components of stress tensor at 1 mL/min by 20 times. In contrast to the flow rate of 1 mL/min, when increasing the flow rate to 8 mL/min, average total shear components of stress tensor became higher and some paths had similar magnitude to average total extensional components of stress tensor and higher by 25 times compared to the flow rate of 1 mL/min (see Figure 6d). As a result, shear stress became significant in the feeding system when increasing the flow rate.

Additionally, the material and length of tubing were considered. The tubing with good biocompatibility, excellent chemical resistance and smooth surface was used to transfer the sample from a syringe to a sorting device. Furthermore, according to simulation of tubing, only shear stress was exerted about 2 Pa and extensional stress was about 0 Pa with an exposure time of 2.43 s. Cell viability, cell morphology and intracellular structures were also examined after flowing through a syringe with two cases—with and without a 20 cm tube. The result showed that tubing had no effect on cell viability and damage. Therefore, the effect of tubing might be ignored in this work.

Again, cell loss was not discovered at either 1 or 8 mL/min. The results demonstrated that in the maximum stress condition (8 mL/min), cells were still viable. Despite the high stresses, the maximum stress area was so small and even smaller than the size of leukocytes at the corner. In other words, cells experienced the maximum stress area with a short exposure time of 0.0125 ms and the paths of most cells were far away from the corner, which could be another important factor that caused the cell not to die. Therefore, the loss of cells was not found in the feeding system in our experiments. However, cell morphology and intracellular structures were slightly damaged as the flow rate increased. It is important to note that a single leukocyte can experience a wide range of stress value depending the path of the streamline in a syringe. It is reasonable to assume that the major contributor causing cell deformation and intracellular damage is the maximum stress area which covers only in a small area at the abrupt change in cross section between syringe’s barrel and attached needle.

### 4.2. The Results of Spiral Microchannel

The examination of cell viability in a complete setup of spiral microchannel, i.e., the feeding system connected with the spiral microchannel, samples were collected in microtubes and measured at only the 1st to 5th outlets. It is noted that an extremely small number of cells came out at the 6th to 10th outlets, resulting in a higher statistical error. Therefore, the 6th to 10th outlets were ignored for cells counting. The data showed that the percentage of cell viability decreased from 95.6% to 85.3% at 2 mL/min and comparably from 95.3% to 88.2% at 10 mL/min. For SEM, the statistics indicated that the percentage of normal cells decreased from 92.2% to 58.6% at 2 mL/min and comparably from 94.2% to 58.3% at 10 mL/min. For Wright-Giemsa stain, we found that the percentage of normal cells were decreased from 95.9% to 91.1% at 2 mL/min and further decreased from 93.6% to 79.3% at 10 mL/min. This implied that cell loss would majorly occur in the spiral microchannel, not in the feeding system.

According to the spiral simulation, it was discovered that, assuming that the cells start moving along the streamline in the middle of the channel, cells then migrated along the streamline in the middle of the channel and moved laterally. After that, they swirled toward the outer walls due to dean vortices on both sides of the upper half and lower half of the channel in the direction of the two counters, rotating clockwise (see Figure 7). While travelling along the streamline, they tended to experience total extensional and shear stress components of the stress tensor. The magnitude of both stresses gradually increased and reached its maximum value. The maximum value of extensional stress was at the position of 2, 3, 5 and 6 (see Figure 7a) on the side walls. Contrarily, the maximum value of the shear stress was at position 1 and 4 (see Figure 7b) on the top and bottom walls. According to the data for the flow rate of 2 mL/min, both total extensional and shear stress components of stress tensor gradually increased from 0 to 18 Pa in the first loop and then gradually decreased from 18 to 0 Pa in the second loop. This movement pattern was repeated continuously until the streamline traveled to the end of fifth loop of spiral due to the secondary flow with a long exposure time of about 0.158 s (see Figure 8a). As a result, the moderate magnitude of total extensional and shear stress components of stress tensor and a long exposure time in spiral had a possibility to cause cell loss and cell deformation during the separation process. This has a good agreement with the past work in Table 1 where the data suggest that low magnitude of shear stress but with long exposure time could cause cell damage in their experiments.

In the case of increasing the flow rate to 10 mL/min, the streamline in the middle of the channel moved toward the outer wall, similar to the condition of 2 mL/min. Then, the streamline quickly moved from the outermost to the innermost wall of the channel in the first loop. This pattern of cell movement traveled more repeatedly and frequently than the condition of 2 mL/min in the same distance of the spiral microchannel. The magnitude of the total extensional and shear components of stress tensor increased from 0 to 300 Pa but the exposure time became shorter (0.032 s). Surprisingly, the magnitude of total extensional components of stress tensor began with 0 Pa at the inlet and went up to 300 Pa at 90° and went down to 0 Pa at 180° (half of a spiral). This happens because the streamline started moving from the middle of the channel and then moved toward the outer wall laterally in the streamwise direction and at the same time, it moved down to the lower half and then up to the upper half of the channel in the spanwise direction. This movement created the trends of the total extensional and shear components of stress tensor that was different from the condition of 2 mL/min. Likewise, the total shear components of stress tensor gradually increased from 0 to 300 Pa and then gradually decreased from 300 to 0 Pa at 360° (see Figure 8b). Interestingly, two flow rates affected cell loss and the morphological change similarly.

For the simulation of the straight channel and 10 outlets, these parts connected to the spiral part as a spiral microchannel. Due to the extremely low magnitude of total extensional and shear components of stress tensor compared to magnitude of extensional and shear stress in the spiral, it is reasonable to assume that the major contributor causing cells to die is the spiral part, while the straight channel and 10 outlets were ignored.

In addition, Figure 8c,d demonstrated that average total extensional and shear components of stress tensor at 2 and 10 mL/min in different locations with nine streamlines were measured by averaging the root sum squared of all extensional or shear components of stress tensor along the nine streamlines. The data showed that the average total shear components of stress tensor was dominant in the spiral microchannel for all streamlines at both 2 and 10 mL/min, with exposure times of about 0.158 s and 0.032 s. Surprisingly, the average total extensional components of stress tensor in the streamline number 1, 4 and 7 at 2 mL/min and the streamline number 1 and 4 at 10 mL/min significantly increased because those streamlines moved towards the outer wall with a long exposure time. However, the average total extensional components of stress tensor of other streamlines was small due to the fact that those streamlines moved away from the walls and stayed close to the center of the channel.

To sum up, the spiral microchannel, either in the optimum condition (lower magnitude of stress with a long exposure time) or the maximum stress condition (higher magnitude of stress with a short exposure time), caused about 10% of cells to die and caused 30% cell deformation in the experiment, similarly. However, shear stress was dominant in the spiral microchannel. The data suggested that the magnitude of shear stress, the pattern of cell movement and long exposure time could be correlated with either cell damage or loss in spiral microchannel. For the damage of intracellular structures, that increased significantly when increasing the flow rate, it might be the results of the frequent occurrence of the event that cells move close to the side wall (high extensional area) at 10 mL/min. Therefore, the possibility that cells experience high extensional stress increased and it caused the damage of intracellular in experiments became significant. 

### 4.3. The Results of Contraction-Expansion Array

The complete setup of CEA showed that cell viability decreased from 96.2% to 89.3% for the main outlet and 88.8% for the secondary outlet at 0.3 and 0.24 mL/min of the main and buffer inlet at optimum condition. SEM data also demonstrated that the percentage of normal cells decreased from 94.9% to 79.6% in the main outlet. For the Wright-Giemsa stain, the percentage of normal cells decreased from 91.0% to 75.6%.

According to the simulation of CEA, the highest extensional and shear components of stress tensor were developed during the abrupt change in cross section between the expansion and contraction region (see Figure 9a). When cells entered this area, cells tended to be stretched suddenly at the corner and then contracted after passing through the stress area, similar to the abrupt change in the feeding system. The contour demonstrated that the total extensional components of stress tensor reached to 550 Pa while shear components of stress tensor was about 700 Pa (see Figure 9b) and exposure time was 0.0125 ms (exposure time was calculated, i.e., 10 µm/0.8 m/s ≈ 0.0125 ms) at the upstream microchamber—10 × 10 µm^2^ (for path 1), while other paths, with an area of 15 × 15 µm^2^ (for path 3 and 4), had lower magnitude of stress compared to the path 1.

To compare with the data of CEA and the feeding system, both data showed that in CEA, the magnitude of total extensional and shear components of stress tensor were higher (550 and 700 Pa at optimum condition in path 1) than the magnitude of stress in the feeding system (500 and 550 Pa at maximum stress condition in path 1), while the maximum stress area of the upstream microchamber was larger (10 × 10 µm^2^) than the area of the syringe’s corner (2.5 × 2.5 µm^2^) compared to leukocytes (7–12 µm) with the same exposure time of 0.0125 ms. Although CEA and the feeding system developed the extensional and shear stress in the same way, extensional and shear stress in CEA exerted more magnitude and a relatively larger area comparing to the entire flow channel than the feeding system in the maximum stress condition. Therefore, higher magnitude of extensional and shear stress and larger high-stress area in CEA could cause more cell damage and loss than the feeding system in comparison.

Furthermore, the data of the Wright-Giemsa stain shows that CEA caused more intracellular damage than the spiral microchannel. This suggests that extensional components of stress tensor in CEA’s microchamber caused cells to be stretched and contracted suddenly, resulting in more intracellular damage than the spiral microchannel. On the other hand, a short exposure time in the CEA’s microchamber would cause less cell deformation due to the shear stress than the spiral microchannel.

According to the data of the average of total extensional and shear components of stress tensor in each path, the magnitude of both were quite similar—350, 325, 250 and 200 Pa, respectively (see Figure 9c). It is reasonable to assume that both extensional and shear stress are dominant equally throughout the streamlines in CEA at optimum condition.

To sum up, the data of feeding system depicted that cells loss was not found in different volume of syringe or increasing flow rates because of a low magnitude of stress with a short exposure time and small high-stress area. However, cell deformation and intracellular damage were slightly found as flow rate increased. For the spiral microchannel, it was discovered that about 10% of cell loss was found in the device as well as about 30% of cell deformation significantly increased because a long exposure time in spiral caused cell to be stretched for a long time leading to cell deformation and eventually caused cell loss. Unlikely, the data of CEA showed that about 7% of cell loss was found while 14% of intracellular damage dramatically increased because a higher magnitude of stresses in CEA’s microchamber with a relatively high-stress area compared to the feeding system caused cell to be stretched and contracted suddenly resulting the disruption of intracellular structures and consequence in cell loss. We summarized the data of stress on cells for each component (see in Table 2).

## 5. Conclusions

According to data of feeding system, the data of cell loss was not found in all cases of the feeding system condition because exposure time is so short at 0.3, 1, 5 and 8 mL/min. Moreover, cell deformation and intracellular damage, membrane, cytoplasm and nucleus break down were slightly found in all cases as well. This was confirmed by the investigation of SEM and Wright-Giemsa stain after passing through the syringe at 0.3, 1, 5 and 8 mL/min.

As for the spiral microchannel, the data demonstrated that cell loss found no significant difference between the flow rates of 2 and 10 mL/min. Although these two flow rates exerted total extensional and shear stress in a different order and pattern of cell’s movement and exposure time due to increasing flow rates. As for SEM, no significant difference between the number of normal cells was found. However, ~30% of cell deformation significantly was increased compared to ~4% in the feeding system and ~12% in CEA. This suggested that total shear components of stress tensor (shear dominated) in spiral microchannel with a long exposure time could cause cells to be sheared along the distance of five-loop spiral resulting in a significant number of cell deformation. As for the experiment of Wright-Giemsa stain, we found that ~2 and ~9% of normal cells were decreased at flow rates of 2 and 10 mL/min. This confirmed that stresses in the spiral microchannel also affect intracellular structures, and the stronger effects could be found at higher flow rates.

On the other hand, the data of CEA showed that the majority of cell loss was found in this part. As for SEM and Wright-Giemsa stain, the percentage of normal cells similarly decreased. Furthermore, the data indicated that CEA exerted higher magnitude of extensional and shear stress and a relatively larger area comparing to the entire flow channel. As a result, the majority of cells that traveled in the larger high-stress area in CEA had a high potential to be damaged in this area. This could cause more cell damage and loss in CEA than the feeding system.

In conclusion, we discovered that the spiral microchannel caused cell deformation rather than CEA. Moreover, shear stress dominated and there was a long exposure time. It is believed that shear stress and a long exposure time in the spiral could cause cells to be sheared gradually and continuously, resulting in nucleus stretching and cell deformation. In contrast, the intracellular damage was occurred more in CEA than in the spiral microchannel where extensional and shear stress dominated. This suggests that at both stresses, components of stress tensor in the upstream microchamber caused cells to be stretched and contracted suddenly, resulting in the disruption of intracellular structures.

## Figures and Tables

**Figure 1 micromachines-10-00772-f001:**
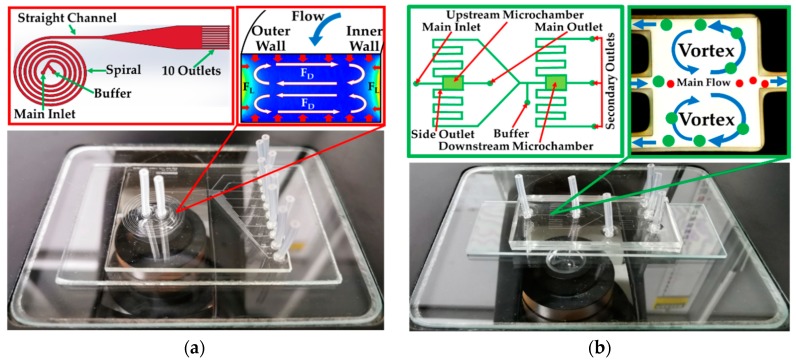
Schematic drawing (not draw to actual scales) and actual device of (**a**) spiral microchannel and (**b**) contraction-expansion array.

**Figure 2 micromachines-10-00772-f002:**
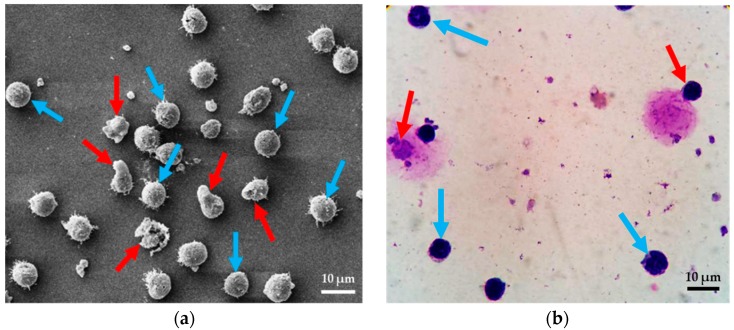
Image of normal cells (blue arrows) and degenerated cells (red arrows) using (**a**) Scanning Electron Microscopy (SEM) and (**b**) Wright-Giemsa stain.

**Figure 3 micromachines-10-00772-f003:**
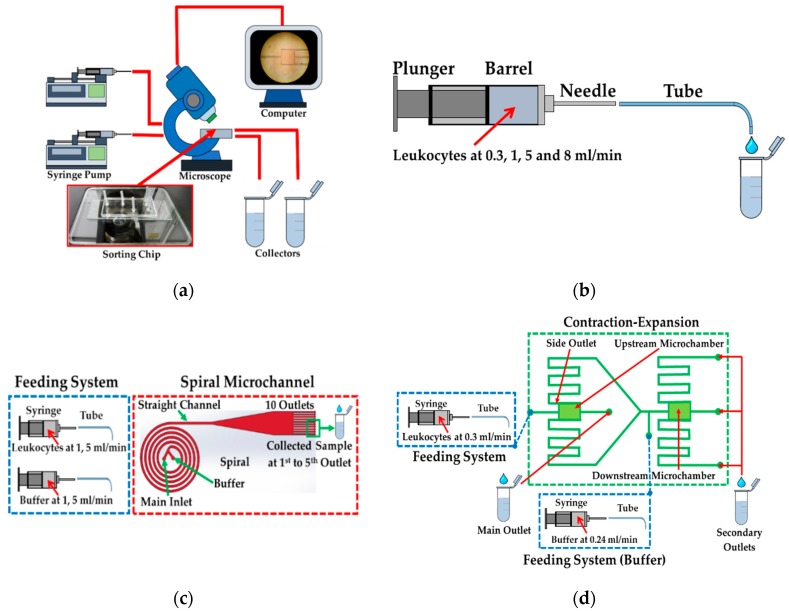
Schematic drawing (not drawn to size) of (**a**) setup of sorting devices, feeding system and spiral microchannel/contraction-expansion array (CEA); (**b**) experiment of feeding system; (**c**) experiment of spiral microchannel and (**d**) experiment of contraction-expansion microchannel.

**Figure 4 micromachines-10-00772-f004:**
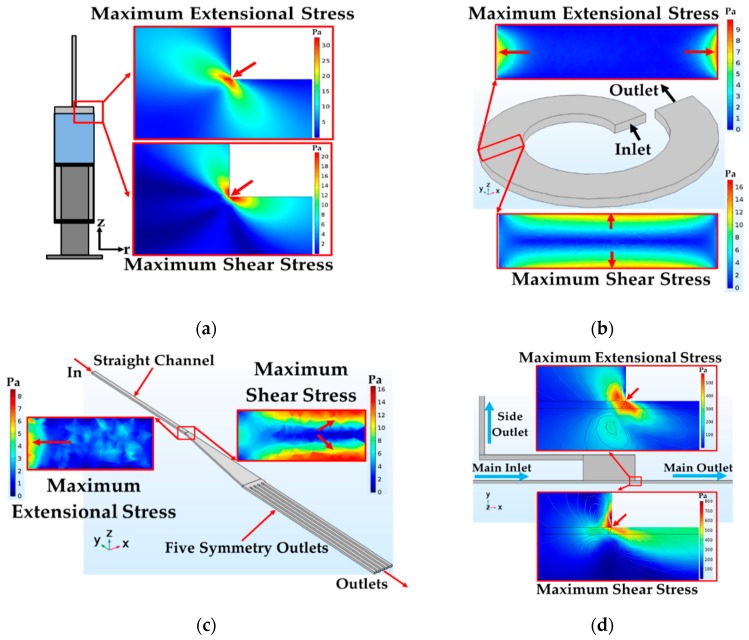
Geometry and the areas of maximum stresses—extensional and shear stress in (**a**) syringe (**b**) spiral microchannel (**c**) straight and 10 outlet sub-channels and (**d**) contraction-expansion array.

**Figure 5 micromachines-10-00772-f005:**
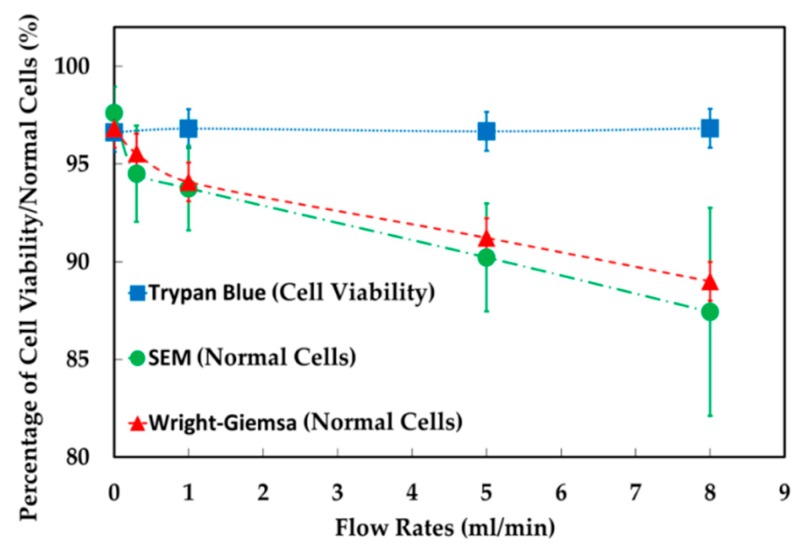
Percentage of cell viability, cell morphology and intracellular structures in feeding system with increasing flow rates.

**Figure 6 micromachines-10-00772-f006:**
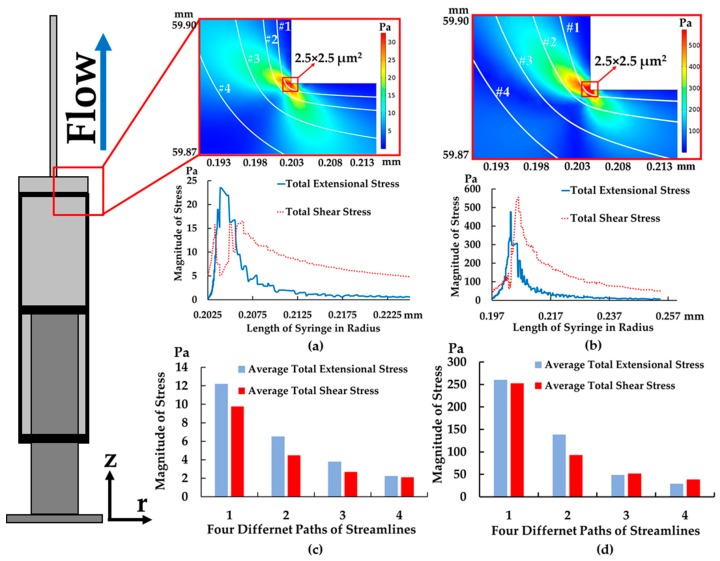
Paths of cell movement with the magnitude of extensional and shear in r-axis: total extensional and shear components of stress tensor (path 1) at (**a**) 1 mL/min (**b**) 8 mL/min and average total extensional and shear components of stress tensor along with four different paths at (**c**) 1 mL/min and (**d**) 8 mL/min (*r* = 0 at the centerline of syringe and *Z* = 0 at the bottom of syringe’s barrel).

**Figure 7 micromachines-10-00772-f007:**
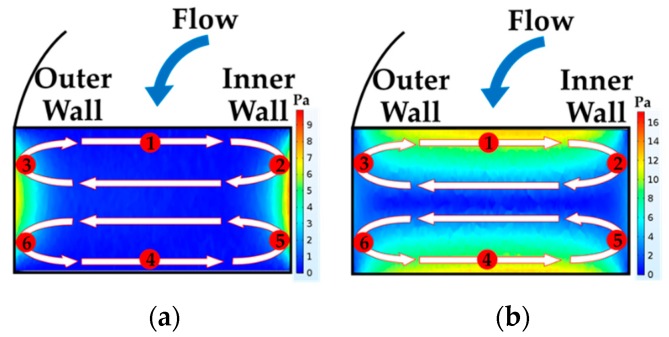
Vortices in a spiral microchannel drive cells travelling in six different locations leading to cell deformation (**a**) extensional stress and (**b**) shear stress at 2 mL/min.

**Figure 8 micromachines-10-00772-f008:**
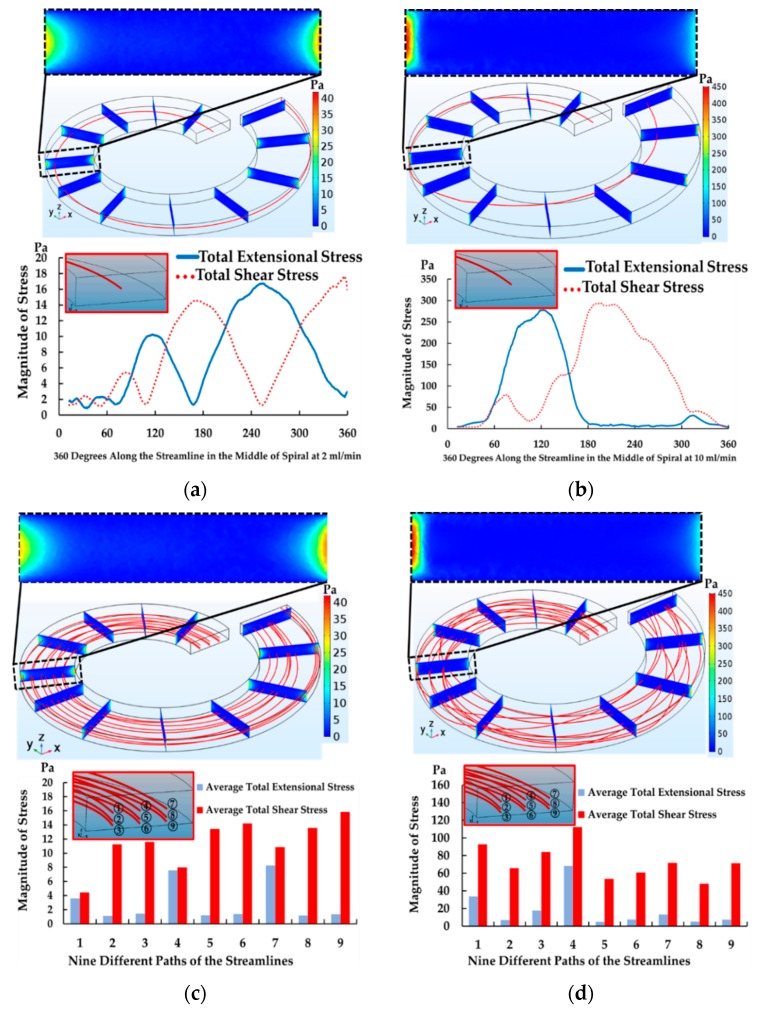
Magnitude of total extensional and shear components of stress tensor (**a**) at 2 mL/min (**b**) at 10 mL/min, along with a streamline in the middle of channel and average total extensional and shear components of stress tensor (**c**) at 2 mL/min and (**d**) at 10 mL/min along with nine different paths of the streamlines in one loop of spiral (0–360°).

**Figure 9 micromachines-10-00772-f009:**
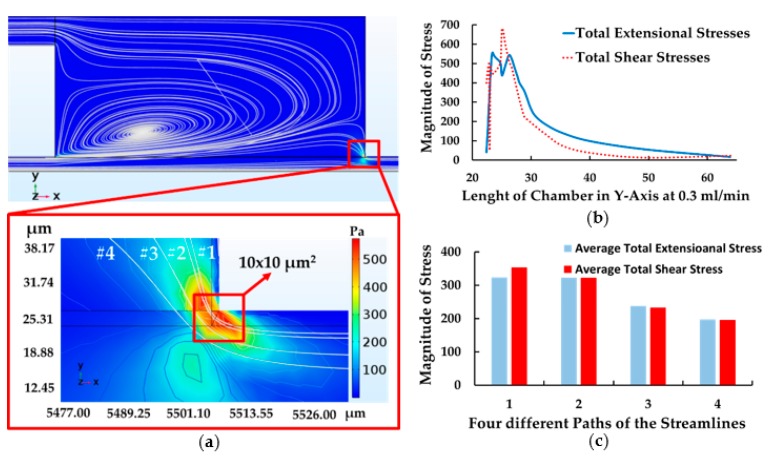
Contraction and expansion chamber at the corner. (**a**) Contour of extensional stress and high stress area, (**b**) the magnitude of total extensional and shear component of stress tensor and the height of chamber in Y-axis (path 1) and (**c**) the magnitude of average total extensional and shear components of stress tensor with four paths of streamlines (Y = 0, at the centerline of the main channel).

**Table 1 micromachines-10-00772-t001:** Summarization of shear and extensional studies over the past decades.

Author	Year	Sample	Method	Type of Stress	Magnitudes of Cell Damage (Pa)	Magnitudes of Cell Loss (Pa)	Time
**Nevaril et al.**	1968	RBCs	Viscometer	Shear	150–300 (Hemolysis)	—	2 min
**Sutera et al.**	1975	RBCs	Viscometer	Shear	<250 (Deformation) >250 (Fragmentation)	—	4 min
**Smith et al.**	1987	Hybridoma	Viscometer	Shear	—	0.67	15 h
**Petersen et al.**	1988	Hybridoma	Viscometer	Shear	—	5	10 min
**Schürch et al.**	1988	Hybridoma	Viscometer	Shear	—	<6	—
**Abu et al.**	1989	Hybridoma	Viscometer	Shear	<5 (Respiration Activity)	>5	0.5–2 h
**Tanzeglock et al.**	2009	HEKs and CHOs	Rheometer and Syringe	Extensional	—	500	—
**Down et al.**	2011	RBCs	Capillary	Extensional	—	3000	Milliseconds
**Aguado et al.**	2012	—	Needle	Extensional	—	—	—
**Yen et al.**	2015	RBCs	Capillary	Extensional	—	1000	0.06 ms
**Bae et al.**	2016	CHOs	Cross-Slot	Extensional	—	250	—

**Table 2 micromachines-10-00772-t002:** Summarization of cell loss and damage for each component.

Component	Flow Rate (mL/min)	Cell Loss (%)	Cell Deformation (%)	Intracellular Damage (%)	Type of Stress	Maximum Stress (Pa)	Time
**Feeding System**	0.3	0	3.1	1.2	Extensional	N/A	N/A
	1	0	3.8	2.7		24	0.25 ms
	5	0	7.4	5.6		N/A	N/A
	8	0	10.2	7.8		500	0.0125 ms
**Spiral**	2	10.3	29.8	2.1	Shear	18	0.158 s
	10	7.1	28.5	8.7		300	0.032 s
**Contraction-Expansion**	0.3 (Main Inlet) 0.24 (Buffer)	6.9 (Main Outlet) 7.4 (Secondary Outlet)	12.2 (Main Outlet) N/A (Secondary Outlet)	14.2 (Main Outlet) N/A (Secondary Outlet)	Extensional and Shear	550 (Extensional) 700 (Shear)	0.0125 ms

N/A is an abbreviation in this tables for the data which are not available.

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
