# Peer review of "Investigation of Leukocyte Viability and Damage in Spiral Microchannel and Contraction-Expansion Array"

_micromachines, 2019, doi:10.3390/mi10110772_

Round 1

Reviewer 1 Report

comments attached.

Reviewer 2 Report

Authors have presented analysis and evaluation of effect of extensional and shear stress on leukocytes in spiral microchannel and expansion and contraction microchannels. The authors have addressed a popular topic of discussion among researchers in the Lab-on-chip field. The effect of shear stress on cells-blood cells, cancer cells, etc has been addressed in previously published work. In fact, a paper published in Micromachine in 2018 addressed the effect of shear stress on cancer cells:Ketpun et al, Micromachine (Basel), 2018. 

The authors have claimed that since there is heterogeniety associated with the type of cell and there are factors such as "experimental approaches, magnitude of stress, types of hydrodynamic stress, exposure time/area," their work holds merit.But, authors forget about this and do not classify their analysis according to it.

Also, the above mentioned work published in 2018 looks very similar to the analysis done by the authors. Just because the cell type is different, it doesn't merit a completely new study unless a drastic experimental need arises. Authors should discuss this. 

In fact, the spiral device design is exactly the same so even the device type is not a factor here.

Authors have not discussed vorticity, high pressure entrapment and toher sources of stress. Sometime during high flow rate regime, vortices form in the expansion channel (right after the spiral channel ends). Please explore and comment.

There has been no discussion of secondary flow and how that interacts with the cell wall and can damage it.

Authors have not provided any quantitative plots quantifying the impact of stress on cells, such as cell viability for each device and each device outlet? extent of damage to cell wall? 

There are no images of cells -damaged or otherwise; stained or bright-field image?

How do authors relate evaluation of the two devices? why only these two devices?

Round 2

Reviewer 1 Report

I appreciate the authors' effects on the manuscript, some of the concerns have been well addressed. Remaining concerns:

1) Experimental validation of the streamline is do-able without the need for any complicated simulation (e.g.: PMID: 24404052). The goal is to make sure the simulated streamlines are real, not artifacts.

2) Since the maximum change of SS always happened at the wall/corner, why not use 2D models instead of 3D? That would save a tremendous amount of computational time.

3) In Fig 4 and 7, the color legend bar indicating SS level (like the ones in Fig 6 and 8) is missing.  

Author Response

I appreciate the authors' effects on the manuscript, some of the concerns have been well addressed. Remaining concerns:

1) Experimental validation of the streamline is do-able without the need for any complicated simulation (e.g.: PMID: 24404052). The goal is to make sure the simulated streamlines are real, not artifacts.

I really appreciate for your suggestion on the streamlines. In fact, another group of our team already studied the streamlines and efficiency of separation device using microbeads (polystyrene beads 5, 10, 15 and 20 microns). However, my study is focused on cell viability and damage of each component. So that, I am afraid that if I put some data or pictures of the streamlines into my work, it will be plagiarized or make another study more difficult to be published

2) Since the maximum change of SS always happened at the wall/corner, why not use 2D models instead of 3D? That would save a tremendous amount of computational time.

            In fact, in spiral microchannel, there is a secondary flow (vortices) that also affect the movement of the streamline (figure). Therefore, if we use 2D instead of 3D model, we would not be able to observe the streamline moving and stress (shear and extensional stress) in cross section. We try to simulate the model as same as an actual situation to see the flow and streamline’s movement.

3) In Fig 4 and 7, the color legend bar indicating SS level (like the ones in Fig 6 and 8) is missing. 

I understand now. I did not put the color legend bar in Fig 4 and 7 because I would like to show only the maximum stress where is located in each device (my intention not to put the color bar). However, if the reviewer think is necessary to add the color legend bar to show more details of stress magnitude, I will add  the bar in both figures. Thank you for your suggestion.

Reviewer 2 Report

Authors have addressed the comments appropriately, however, there are a few points that still need to be elaborated.

Spiral and CEA are widely studied and even the effect of tubing has been studied. It is not necessary that the method of infusion is same as authors'. The shear and flow stress could be due to the material of the tubing as well (how stiff the tubing is: Peek,FEP,PTFE, etc). Authors should evaluate and discuss that. The length of the tubing vs the flow rate is also an important parameter.

Although authors have tried to summarize the quantitative data in table (table 2), it would be insightful to include some representative plots-especially plots of % cell loss vs the flow rate/infusion rate? -each for feeding system, spiral and CEA.

Authors are requested to proof read the document carefully. There are a number of run-on sentences and grammatical errors.

Author Response

Authors have addressed the comments appropriately, however, there are a few points that still need to be elaborated.

1.Spiral and CEA are widely studied and even the effect of tubing has been studied. It is not necessary that the method of infusion is same as authors'. The shear and flow stress could be due to the material of the tubing as well (how stiff the tubing is: Peek,FEP,PTFE, etc). Authors should evaluate and discuss that. The length of the tubing vs the flow rate is also an important parameter.

I really appreciate for your suggestion on the evaluation of the tubing and material, length of tubing and flow rate. Once the material and length of tubing were also considered. The tubing with good biocompatibility, excellent chemical resistance and smooth surface was used to transfer the sample from a syringe to a sorting device. Furthermore, according to simulation of tubing (Figure), only shear stress was exerted about 2 Pa and extensional stress was about 0 Pa with exposure time of 2.43 s. Cell viability, cell morphology and intracellular structures were also examined after flowing through a syringe with two cases—with and without a 20 cm tube. The result showed that tubing had no effect on cell viability and damage. Therefore, the effect of tubing might be ignored in this work. Similar to a needle, needle provides the data in the same way (only shear stress exerted) but the needle is a part of a syringe (cannot be evaluated separately).

Thank you for your suggestions, I will add these in my manuscript.

2.Although authors have tried to summarize the quantitative data in table (table 2), it would be insightful to include some representative plots-especially plots of % cell loss vs the flow rate/infusion rate? -each for feeding system, spiral and CEA.

I will fix and summarize the quantitative data including % cell loss and flow rate for feeding system, spiral and CEA to make a better understanding of cell damage for each device.

3.Authors are requested to proof read the document carefully. There are a number of run-on sentences and grammatical errors.

Thank you so much and appreciate it.
